# Preoperative Spinal Angiography for Thoracic Neuroblastoma: Impact of Identification of the Adamkiewicz Artery on Gross Total Resection and Neurological Sequelae

**DOI:** 10.3390/children10071116

**Published:** 2023-06-27

**Authors:** Angelo Zarfati, Cristina Martucci, Giorgio Persano, Giulia Cassanelli, Alessandro Crocoli, Silvia Madafferi, Gian Luigi Natali, Maria Antonietta De Ioris, Alessandro Inserra

**Affiliations:** 1General and Thoracic Pediatric Surgery Unit, Bambino Gesù Children’s Hospital, IRCCS, 00165 Rome, Italy; 2Surgical Oncology Unit, Bambino Gesù Children’s Hospital, IRCCS, 00165 Rome, Italy; 3Interventional Radiology Unit, Bambino Gesù Children’s Hospital, IRCCS, 00165 Rome, Italy; 4Department of Hematology/Oncology, Cell Therapy, Gene Therapies and Hemopoietic Transplant, Bambino Gesù Children’s Hospital, IRCCS, 00165 Rome, Italy

**Keywords:** angiography, neuroblastoma, mediastinal tumors, children, pediatric oncology

## Abstract

**Simple Summary:**

Patients with thoracic neuroblastoma are at high risk of operative neurologic complications due to iatrogenic lesions of the artery of Adamkiewicz (AKA), reducing the possibilities of gross total resection. Therefore, the role of preoperative spinal angiography in such patients must be defined. This study aimed to investigate the role of spinal angiography in thoracic neuroblastoma and the impact of accurately identifying the AKA on surgical resection and neurological complications.

**Abstract:**

Background: Patients with thoracic neuroblastoma (TNB) are at high risk of postoperative neurologic complications due to iatrogenic lesions of the artery of Adamkiewicz (AKA). The role of performing a preoperative spinal angiography (POSA) in these patients must be clarified. The present study sought to further understand the relationship between POSA and TNB, as well as the effects of identifying the AKA on surgical excision and neurological consequences. Methods: Data from patients with TNB who underwent POSA between November 2015 and February 2022 at our tertiary pediatric center were retrospectively analyzed. Results: Six patients were identified, five of whom (83%) were considered eligible for surgical excision. Gross total resection (GTR) was achieved in three patients (60%), which included two patients with an AKA contralateral to the tumor, and one with an homolateral AKAl. After a median follow-up of 4.1 years from diagnosis, no patients developed neurological complications; five (83%) were alive and well, and one died from refractory recurrence. Conclusions: Among patients with TNB, POSA was useful for identifying the AKA and defining the optimal surgical strategy. POSA should be considered in the preoperative evaluation of TNB to increase the likelihood of GTR and reduce the threats of iatrogenic neurologic sequelae.

## 1. Introduction

The most frequent solid extracranial tumor in children is neuroblastoma [1], which develops from primitive neural crest cells and can arise in the adrenal medulla and anywhere along the sympathetic nervous system [2]. The presentation and clinical features of neuroblastoma may be extremely heterogeneous [3]. Approximately 15–25% of neuroblastoma occur in the posterior thorax, where intraspinal extension is relatively common [2,3,4]. Posterior thoracic tumors may involve the origin of the Adamkiewicz artery (AKA), with an inherent iatrogenic risk of vascular damage and consequent spinal cord ischemia [5,6,7,8]. The AKA contributes to the vascularization of the anterior spinal artery and arises from the intercostal artery, usually between thoracic vertebral levels T9 and T12 [9]. However, the origin of the AKA may vary, and has been described as arising anywhere between T5 and L2 [8,9]. Due to the possible operative risk of AKA injury, infiltration between T9 and T12 (costovertebral junction) is considered an image-defined risk factor (IDRF) for neuroblastoma [8,9]. This localization reduces the chances of gross total resection (GTR) and increases the risk of neurologic complications and sequelae [6]. GTR plays a crucial role in local control and the overall survival of patients with neuroblastoma [10,11,12,13]. Therefore, the surgical management of patients with thoracic neuroblastoma (TNB) is challenging, even in experienced centers. Few studies have proposed the use of preoperative spinal arteriography (POSA) for assessing patients with various posterior mediastinal tumors to limit morbidity and increase the possibilities of complete resection [6,7,8]. Such investigations are invasive and associated with an increased risk of severe procedure-related complications [14]. Therefore, the utility of POSA in the preoperative assessment of TNB remains unclear.

This study aimed to investigate the role of POSA in TNB and the impact of identifying the AKA on GTR, neurological complications, and sequelae.

## 2. Materials and Methods

This single-center retrospective study included consecutive pediatric patients with TNB who underwent POSA at our center between November 2015 and February 2022. The indication for POSA was based on the diagnosis of TNB extending into the costovertebral junction. Depending on the disease stage, age of the patients, and histological findings, patients underwent preoperative chemotherapy according to the treatment guidelines of the Low and Intermediate Risk Neuroblastoma European Study (LINES) or the European SIOP Neuroblastoma Group (SIOPEN HR-NBL 1).

The clinical, radiological, and surgical information, follow-up data, and outcomes were reviewed. The primary outcomes evaluated included periprocedural and perioperative findings and complications, GTR, and long-term neurological sequelae. According to international definitions [10,12,13], GTR was defined as ≥90% reduction, measured according to the intraoperative surgeon’s assessment and postoperative computed tomography (CT) findings.

A preoperative transfemoral angiography using 4 Fr catheter was performed in all patients included in the study. Selective intercostal left and right digital subtractive angiography was achieved in posterior–anterior projection to visualize and assess the origin of the AKA, usually originating from the ninth left intercostal artery but often presenting with anatomical variants.

## 3. Results

Ninety-two patients with neuroblastoma were treated between November 2015 and February 2022 in our Institution, but only six met the eligibility criteria (thoracic neuroblastoma with an extension in the costovertebral junction and POSA performed). Baseline characteristics and patient diagnoses are summarized in Table 1. The median age at diagnosis was 2.7 years (range: 0.5–16.4 years) and three patients (50%) were female. The mass was on the left side in five patients (83%) and located thoracoabdominal in three cases (50%). All patients had at least one IDRF, with a median of 2 (range: 1–8 IDRFs). The median maximal diameter of the mass at diagnosis was 12 cm (range: 6–20 cm). Two patients (33%) had MYC-N amplification. Three patients (50%) were classified as high risk at diagnosis (and treated according to NB-HR-O1) and the other three as intermediate risk (and treated according to the LINES protocol).

Periprocedural information and the spinal angiography results are summarized in Table 2. The median age at spinal angiography was 2.3 years (range: 1.5–16.6 years). POSA was performed after a median of 5 months after diagnosis (range: 3–30 months). The patients exhibited a median of 2.5 IDRFs (range: 1–6) before the spinal angiography. The AKA was always identified as emerging from an intercostal artery. In five patients (83%), POSA revealed the AKA on the left; in two (33%), the AKA was contralateral to the tumor, whereas in the remaining four (66%), the AKA was homolateral to the mass. In the latter group, one patient had the origin of the AKA at the level of T11, with the tumor extending from T3 to T9, and in the remaining three, the origin of the AKA was encased by the tumor [Figure 1]. No periprocedural complications were observed. Five patients (83%) underwent surgical excision after POSA, while one patient was excluded from resection based on the extension of the disease.

Surgical details, follow-up, and long-term outcomes are summarized in Table 3. Surgical access was through posterolateral thoracotomy in four patients (80%), and a median sternotomy was performed in one patient (20%). Intraoperative complications occurred in two patients (40%): lung laceration in one, and aortic lesion in the other (due to tenacious adhesion of the mass to the aortic origin). Both lesions were recognized intraoperatively and treated by primary suture with no postoperative sequelae. GTR was possible in three patients (60%). Among them, GTR was feasible in two patients (100%) with the AKA contralateral to the tumor, and GTR was possible in one patient (1/3; 33%) in whom the AKA originated on the same side of the tumor but was not encased by it.

The patients were followed up radiologically and clinically, every 2–4 weeks; the median follow-up was 49 months from diagnosis (range: 31–96 months). Two patients (33%) experienced a distant metastatic relapse and no patients presented neurological sequelae after surgery or during the follow-up. Five patients (83%) were alive and well at the end of the follow-up, while one died after a refractory recurrence.

## 4. Discussion

Spinal cord vascular injury due to iatrogenic damage of the AKA is a rare but severe complication of posterior thoracic tumors [6,7,8,15], which can lead to severe and permanent neurological sequelae [5,8]. Studies investigating spinal arterial vascularization have been advocated to reduce the risk of this unfortunate event [5,8].

There is some utility for the preoperative identification of the AKA using CT or magnetic resonance imaging (MRI) [9], with adequate visualization of the artery in 95% and 93% of patients, respectively [16,17]. In particular, CT angiography is effective in depicting the AKA in patients as young as 5 years of age with a high degree of sensitivity [17]. However, one study comparing traditional arteriography with CT angiography reported that arteriography was more effective than CT in identifying the AKA (94% vs. 60%) and establishing continuity (87% vs. 56%) [18]. Thus, spinal angiography currently appears to be more accurate in identifying and tracing the AKA and is considered to be the gold standard for pediatric patients [6,8].

Due to the risk of severe procedure-related complications [14], indications for POSA must be selected carefully. At our institution, the procedure is reserved for patients with TNB extending into the costovertebral junction, regardless of the lesion level.

In fact, even if the vertebral levels T9–T12 are the boundaries in the International Neuroblastoma Risk Group Staging System (INRGSS) for IDRFs concerning infiltration of the costovertebral junction, a recent meta-analysis reported that 7.3% and 6.9% of AKAs arise at T8 and L1, respectively [19]. Furthermore, in an autopsy series of 51 cadavers, Biglioli et al. [20] described the AKA located below T12 in 70.5% of cases, between L1 and L3 in 64.7% of cases, and below L2 in 23.5%, whereas Lo et al. reported 3 cases of L4 AKAs in a wide angiographic series [21]. In approximately 70% of cases (67.7–83%), the AKA arises from the left-sided intercostal or lumbar artery [22].

In patients with TNB, defining the anatomy of the AKA plays a crucial role in surgical planning. In fact, the ability to GTR while preserving vascularization of the spinal cord has a favorable impact on local control of the disease and general outcomes in patients with neuroblastoma, including those at high risk [10,11,12,13]. However, when the AKA is clearly identified, it must be carefully preserved, even if the procedure is a nonradical excision. Otherwise, when the AKA is absent and several roots must be resected, it is crucial that contralateral corresponding intercostal arteries are present and strictly preserved.

We presented our pilot experience regarding the utility of POSA for GTR of TNB. In the present case series, spinal angiography was safe for experienced surgeons, and no periprocedural complications occurred. POSA effectively identified the origin of the AKA in all patients and helped tailor the surgical approach, thus minimizing neurological complications and their sequelae. In fact, patients in whom the origin of the AKA was found to be contralateral to the mass (2/6) or homolateral but not encased by the tumor (1/6) underwent an aggressive but safe surgical approach, achieving GTR without the risk of spinal cord injury. None of the patients developed neurologic complications or long-term sequelae, both perioperatively and at follow-up.

Although all previous studies used catheter-based angiography, where catheterization techniques and the extent of angiography varied, the results of the present study are consistent with those in the recent literature [23]. Champlin et al. reported that the AKA was clearly recognized in 36% of patients when catheterized two levels above and below the intended surgical site [24]. More extensive angiography, performed by Fanous et al. and Charles et al., led to the successful identification of the AKA in 71% and 96% of patients, respectively, demonstrating that, with wider angiography, there was a higher possibility of detection [25,26]. Furthermore, in 2008, a study by Uotani et al., involving 32 patients with thoracic or thoracoabdominal aortic aneurysms, demonstrated that intra-arterial contrast injection was significantly more sensitive in identifying the AKA compared to intravenous contrast injection (94.1% vs. 60%), suggesting that this technique is the gold standard for spinal angiography [18].

As previously reported [8], POSA is safe and effective, with modest radiation and contrast doses in children, despite the risk of increased radiation exposure, intravenous contrast dose, and procedural problems. The existing literature has reported some risks after spinal angiography, such as retroperitoneal bleeding, temporary cerebral ischemia, transient paresis of the lower extremities, and spinal ischemia in approximately 4.6% of patients [27]. However, these rates are based on earlier imaging protocols in which hyperosmotic ionic contrast agents were used. A recent study using nonionic contrast with short catheterizations and frequent heparin flushes reported the risk of neurologic deficits, contrast-induced nephrotoxicity, and access site complications to be ≤0.8%, ≤0.4%, and ≤1.0%, respectively [28].

However, no access site- or contrast-induced complications were observed in our cohort. Moreover, the mitigated risk of potentially permanent paresis or paraplegia in the absence of POSA outweighs the risks associated with angiography. Furthermore, a more informed discussion of risks and benefits when assessing treatment options with patients and caregivers is made possible by the information provided by POSA.

The study has some obvious limitations due to its single-center retrospective design. The small sample size (a function of the low incidence of TNB) and lack of randomization also limit the generalization of the results. Furthermore, children with TNB represent a small population with peculiar characteristics and we acknowledge that POSA may not be available in developing countries or lower resourced settings, but we strongly believe that the treatment of these patients (managed by various professional health personnel) can benefit from this radiological technological advancement. Therefore, future prospective, multicenter, randomized studies are warranted to confirm our findings.

## 5. Conclusions

Among children with TNB, POSA was a safe and effective procedure for identifying the AKA and defining the optimal surgical strategy in an experienced center. Spinal angiography should be considered for the preoperative evaluation of patients with TNB to maximize the possibility of GTR and reduce the risk of iatrogenic damage to the AKA and its neurological consequences.

## Figures and Tables

**Figure 1 children-10-01116-f001:**
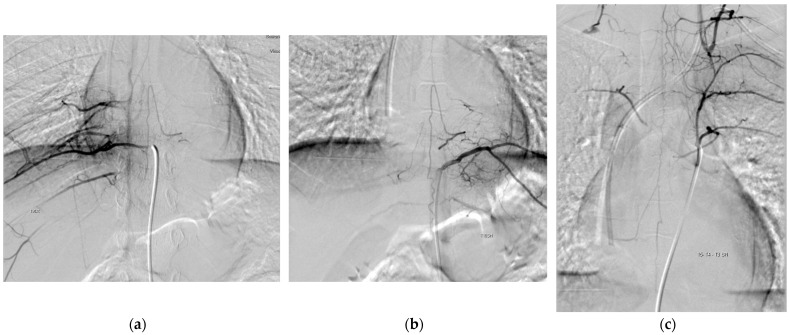
(**a**) PA Right Anterior Oblique projection (10°) selective angiography of right T9 metameric artery demonstrates vascular afferences to inferior and superior branch of AKA. (**b**) PA Right Anterior Oblique projection (10°) selective angiography of left T10 metameric artery demonstrates vascular afferences to inferior and superior branch of AKA. (**c**) PA Left Anterior Oblique projection (LAO) (10°) selective angiography of left T5 metameric artery demonstrates anastomotic vessels between T3 to T5 with evidence of vascular afferences mostly to superior branch of AKA, in its thoracic component.

**Table 1 children-10-01116-t001:** Clinical and oncological characteristics of the analyzed population.

Patient Nr	Sex	Age at Diagnosis (y)	Side	Thoracoabdominal	Number of IDRFs	n-MYC Amplified	D Max (cm)	Stage Risk	Protocol
1	M	16.4	L	No	1	Yes	18 × 8.5 × 8	High (M)	COJEC
2	F	1.9	R	No	1	No	3.5 × 5.1 × 6	Intermediate (L2)	LINES
3	F	0.5	L	Yes	8	No	5 × 3.8 × 7	Intermediate (L2)	LINES
4	M	3.5	L	No	2	Yes	14 × 6 × 6.6	High (M)	COJEC
5	F	1.7	L	Yes	4	No	5 × 5.5 × 10	Intermediate (L2)	LINES
6	M	3.9	L	Yes	DU	No	11 × 7 × 20	High (M)	COJEC

IDRF: image-defined risk factors; DU: data unknown.

**Table 2 children-10-01116-t002:** Radiological data of the preoperative Spinal Angiographies performed.

Patient Nr	Diagnosis–POSA (Months)	Nr IDRFs Pre-POSA	Age at POSA (y)	Complications	AKA Origin	AKA-TNB Position	Management Post-POSA
1	4	1	16,6	No	Left intercostal artery T11	Homolateral	Surgery
2	4	1	2,2	No	Left intercostal artery T5	Contralateral	Surgery
3	13	4	1,6	No	Left intercostal artery T10	Homolateral	No surgery
4	3	2	3,7	No	Left intercostal artery T10	Homolateral	Surgery
5	6	3	2,2	No	Right intercostal artery T9	Contralateral	Surgery
6	30	6	1,5	No	Left intercostal artery T9	Homolateral	Surgery

SA: Spinal Angiography; IDRF: image-defined risk factors; AKA: Adamkiewicz artery, TNB: Thoracic Neuroblastoma.

**Table 3 children-10-01116-t003:** Surgery, follow-up, and outcomes of the analyzed population.

Patient Nr	Access	Intraoperative Lesions	Gross Total Resection	FU Since Diagnosis (Years)	Relapse	Neurologic Complications	Status at Last FU
1	Median sternotomy	No	Yes	3.9	Yes	No	Alive and well
2	Posterolateral thoracotomy	Lung	Yes	3.6	No	No	Alive and well
3	No surgery	NS	NS	8.0	No	No	Alive and well
4	Posterolateral thoracotomy	No	No	2.6	No	No	Alive and well
5	Posterolateral thoracotomy	No	Yes	4.3	No	No	Alive and well
6	Posterolateral thoracotomy	Aorta	No	6.5	Yes	No	Deceased

FU: follow-up; NS: no surgery.

## Data Availability

The raw data supporting the conclusions of this article will be made available by the authors, without undue reservation.

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
