# Peer review of "Preoperative Spinal Angiography for Thoracic Neuroblastoma: Impact of Identification of the Adamkiewicz Artery on Gross Total Resection and Neurological Sequelae"

_children, 2023, doi:10.3390/children10071116_

Round 1

Reviewer 1 Report

It is an interesting study but with many gaps:

Regarding the composition:

Citations at the end of the sentence

Line 66 – our was. Please complete

Line 80 – clarify A 4 Fr

Line 84 – nineth

Line 143 – after the brackets is a 1 that must be removed

In terms of substance:

The material and method are not very legible, this aspect must be reviewed

How many patients did not meet the eligibility criteria?

A short summary of the LINES and NB-HR-O1 protocols should be added in the material and method

In the tables, try to enter the age at the time of diagnosis in years and months, and for the statistical study preferably months

Is reference 15 only for self-citation?

The most important aspects are related to the nature of the type of article, it describes a series of cases, so it must be classified as a case series.

How was the follow-up of the patients carried out? A follow-up of one year is insufficient.

The study group is actually much too small, even if the incidence of these tumors is low.

Aspects related to neuroblastoma are neglected.

The title is not sufficiently represented in the article. The emphasis was placed on the risk of the surgical procedure and the idea of neurological sequelae was neglected. In these aspects, the fact that it is still of a tumor nature (which was neglected in the discussions) must also be taken into account.

Lines 166-169 – have no references

Line 182 – to rewrite the reference

In terms of substance:

The material and method are not very legible, this aspect must be reviewed

How many patients did not meet the eligibility criteria?

A short summary of the LINES and NB-HR-O1 protocols should be added in the material and method

In the tables, try to enter the age at the time of diagnosis in years and months, and for the statistical study preferably months

Is reference 15 only for self-citation?

The most important aspects are related to the nature of the type of article, it describes a series of cases, so it must be classified as a case series.

How was the follow-up of the patients carried out? A follow-up of one year is insufficient.

The study group is actually much too small, even if the incidence of these tumors is low.

Aspects related to neuroblastoma are neglected.

The title is not sufficiently represented in the article. The emphasis was placed on the risk of the surgical procedure and the idea of neurological sequelae was neglected. In these aspects, the fact that it is still of a tumor nature (which was neglected in the discussions) must also be taken into account.

The opinion of the ethics commission is not found.

Author Response

It is an interesting study but with many gaps

Thank you for your valuable comments!

Citations at the end of the sentence

Whenever possible, we reported the citation at the end of the sentence, excluded the ones referring to a specific author/topic cited in text.

Line 66 – our was. Please complete

Line 80 – clarify A 4 Fr

Line 84 – nineth

Line 143 – after the brackets is a 1 that must be removed

A short summary of the LINES and NB-HR-O1 protocols should be added in the material and method

We corrected the text according to your suggestions

Lines 166-169 – have no references

Ref 10-13 are already reported at the end of this paragraph.

Line 182 – to rewrite the reference

Data related to our population do not require any reference, in our opinion.

The material and method are not very legible, this aspect must be reviewed

The manuscript underwent a complete language revision

How many patients did not meet the eligibility criteria?

Ninety-two patients with Neuroblastoma were treated between November 2015 and February 2022 in our Institution, but only six met the eligibility criteria (Thoracic Neuroblastoma with an extension in the costovertebral junction and POSA performed).

In the tables, try to enter the age at the time of diagnosis in years and months, and for the statistical study preferably months

Due to the small size of the tables and the age variability of the patients analyzed, we kept the age in the tables in years, while we changed the ages in months in the results as per your indications

Is reference 15 only for self-citation?

Ref 15 has been cited because it represents one of the few case series of patients reported in the literature describing post-operative paraplegia due to injury of Adamkiewicz artery.

The most important aspects are related to the nature of the type of article, it describes a series of cases, so it must be classified as a case series.

Following the scheme of a case series, we reported data of a small group of patients with same diagnosis and management, treated at our Institution. We summarized their features in the results and specified single patient’s characteristics in the tables. Unfortunately, Children doesn’t have “Case Series” as Article Type, but only Case Report or Article and the latter, in our opinion, seems to represent better our original research.

How was the follow-up of the patients carried out? A follow-up of one year is insufficient.

The patients were followed up radiologically and clinically, every 2-4 weeks. At the time of data collection (in December 2022), the median follow-up was 4.1 years from diagnosis, ranging from 2.6 to 8 years; however, except for the deceased one, all the patients are still under follow-up at our institution.

The study group is actually much too small, even if the incidence of these tumors is low.

We realize that the group of patients analyzed is very small, however we strongly believe that they represent a rare, yet interesting population. This study is the first ever analyzing the use of Pre-operative Angiography in Thoracic Neuroblastoma, correlating radiological features with neurological outcome. 

Aspects related to neuroblastoma are neglected. The title is not sufficiently represented in the article. The emphasis was placed on the risk of the surgical procedure and the idea of neurological sequelae was neglected. In these aspects, the fact that it is still of a tumor nature (which was neglected in the discussions) must also be taken into account.

As previously reported in the literature (also in our Institution), the accidental lesion of Adamkiewicz artery (with consequent spinal ischemia) leads to disabling neurological sequelae. To date, NBL treatment protocols do not recommend mutilating surgery, which surgery with neurological complications would be. In addition, the angiographic study, in cases where it identifies the emergence of AKA at a level not involved by the tumor, transforms an L2 tumor into an L1 tumor, which justifies a more aggressive and in any case non-mutilating treatment strategy. Thus, we strongly believe that angiography can be a useful tool in the surgical planning of patients with neuroblastoma who, despite having a malignant tumor, must not be burdened by neurological consequences as a result of the procedure.

The opinion of the ethics commission is not found.

Following the policy of our Institution, due to the retrospective nature of the study, Ethical Board Approval was not necessary; however, the manuscript received authorization for publication from the Institutional Scientific board before submission.

Reviewer 2 Report

This report is a retrospective study for thoracic neuroblastoma who underwent preoperative angiography to identify Adamkiewicz artery. The number of patients is very small, but there are some important information for pediatric oncologic surgeons.

I have a few questions.

1. Table1 'c-MYC amplified' is 'n-MYC'?

2. Please descrive the type of relapse. local reccurence? distant metastasis?

Author Response

This report is a retrospective study for thoracic neuroblastoma who underwent preoperative angiography to identify Adamkiewicz artery. The number of patients is very small, but there are some important information for pediatric oncologic surgeons.

Thank you for your review. We strongly believe that our study, albeit small, could suggest POSA as a useful “tool” for the management of patients with neuroblastoma.

I have a few questions.

    1. Table1 'c-MYC amplified' is 'n-MYC'?

Yes: we corrected the table.

  1. Please describe the type of relapse. local recurrence? distant metastasis?

We add this information about relapse in the “Results” section.

Reviewer 3 Report

The artticle is a case series on thoracic neuroblastoma and angiography to identify the AKA.

Language:

-        Language in the second half of the article is of good quality, but the abstract and first half need to be reviewed for grammar.

-          L66: “Our was…” is the word “aim” missing?

-          L 70 and L208: “Not monocentric” but “single center”

-          L73: correct the grammar please.

-          L76: not “revised” but “reviewed”

-          L78: not “previous” but “standardize” of “international”

-          L83: not “endowed” but “presenting”

-          L167-169: Please review grammar.

Plagarism:

The opening line of the introduction (L44-46) has been published in several articles. This may border on plagiarism. The reviewer suggests a revision of the opening sentence.

Content:

-          What do the authors mean by L151 “indications for POSA must be selected”? Do they mean that doing the procedure must be done selective or that the indications should still be established?

-          The authors have not sufficiently contrasted spinal angiography with other options. An two phase MRI angio is an alternative, though less sensitive. This is an important discussion as not all centers are equally resourced and spinal angiography isn’t readily available.

-          The authors argue that the risk benefit ratio for doing spinal angiography is acceptable, yet approximately 10% of patients have paraspinal NB and less than 25% are of posterior thoracic NB origin. Of these less than 8% of the patients had encased AKA. Thus, you have to do between 11-15 angiographies to identify one encased of aberrant anatomical placement. Which according to the literature has a less than 5% complication rate. The reviewer feels that the argument to promote spinal angiography as gold standard has not been convincingly presented.

-          A further consideration is the survival rates in HR and IR disease in relation to the procedure.

-          Although this is an interesting case the discussion for this case is limited and could benefit with more nuances. This article is also very specialized for a general paediatric journal. The reviewer suggests a specialised radiography or surgery journal.    

Tables: Acceptable

-          Table 2: please also include TNB – thoracic neuroblastoma in the abbreviations

Figures: Acceptable

Author Response

The article is a case series on thoracic neuroblastoma and angiography to identify the AKA.

Thank you for your useful suggestions!

Language in the second half of the article is of good quality, but the abstract and first half need to be reviewed for grammar.

L73: correct the grammar please.

L167-169: Please review grammar.

The manuscript underwent a complete language revision

L66: “Our was…” is the word “aim” missing?

L 70 and L208: “Not monocentric” but “single center”

 L76: not “revised” but “reviewed”

L78: not “previous” but “standardize” of “international”

L83: not “endowed” but “presenting”

Table 2: please also include TNB – thoracic neuroblastoma in the abbreviations

The opening line of the introduction (L44-46) has been published in several articles. This may border on plagiarism. The reviewer suggests a revision of the opening sentence.

We corrected the text according to your suggestions

What do the authors mean by L151 “indications for POSA must be selected”? Do they mean that doing the procedure must be done selective or that the indications should still be established?

We believe that, due to the risk of possible complications related to the procedure, POSA should be performed in selected patients (Thoracic Neuroblastoma with extension in the costovertebral junction) whose radiological characteristics may help the surgeon to achieve a more complete resection of the tumor without injuring the Adamkiewicz artery.

The authors have not sufficiently contrasted spinal angiography with other options. An two phase MRI angio is an alternative, though less sensitive. This is an important discussion as not all centers are equally resourced and spinal angiography isn’t readily available.

In our opinion, there is no exam comparable to Spinal Angiography in identifying and tracing the AKA; in the discussion we mentioned the use of MRI or CT but, as other authors suggest, Spinal Angiography is nowadays considered the gold standard, even in pediatric patients. However, we realize that this technique is not so widely diffuse and should be performed by very skilled radiologists, in order to reduce possible complications. In low-resourced hospitals, other radiological exams could be performed, if referring the patients to other centers is not possible.

The authors argue that the risk benefit ratio for doing spinal angiography is acceptable, yet approximately 10% of patients have paraspinal NB and less than 25% are of posterior thoracic NB origin. Of these less than 8% of the patients had encased AKA. Thus, you have to do between 11-15 angiographies to identify one encased of aberrant anatomical placement. Which according to the literature has a less than 5% complication rate. The reviewer feels that the argument to promote spinal angiography as gold standard has not been convincingly presented.

Accidental lesions of the Adamkiewicz artery, resulting in spinal ischemia, although rare, have previously been documented in the literature (and in our institution as well). NBL treatment guidelines do not currently support mutilating surgery, which would entail surgery with neurological repercussions. Additionally, the angiographic examination converts an L2 tumor into an L1 tumor when it detects the formation of AKA at a level unrelated to the tumor, which warrants a more aggressive and, in any event,non-mutilating therapeutic approach. Therefore, despite having a malignant tumor, patients with neuroblastoma should not be plagued by neurological complications as a result of the treatment, and we firmly believe that angiography can be a beneficial tool in surgical planning.

A further consideration is the survival rates in HR and IR disease in relation to the procedure.

The ability to perform a complete resection of the tumor positively affects the prognosis of patients with neuroblastoma, whether they have HR or IR; the use of angiography, which permits in some cases more radical (but still safe) interventions, could therefore positively affect the prognosis of these children.

Although this is an interesting case the discussion for this case is limited and could benefit with more nuances. This article is also very specialized for a general paediatric journal. The reviewer suggests a specialised radiography or surgery journal.    

Since the management of patients with neuroblastoma involves various professional figures, from oncologists to pediatricians, from surgeons to radiologists, we wanted this study to reach a wider audience, not limiting it to a highly specialized area of ​​interest.

Reviewer 4 Report

A study evaluating a method to improve the safety of surgery in neuroblastoma is a very reasonable study to perform, as there is significant potential for morbidity. However, I find it difficult to draw significant conclusions related to the long term survival outcomes presented in this study due to the variability in the underlying biology of the tumors. A report of 3 intermediate risk children and 3 high risk children is difficult for me to interpret. It might be helpful if there is a comparison group that can be provided to show the rates of neurologic injury without angiography being performed. It is clear that spinal angiography is possible and that it identifies the origin of the AKA, but I'm not sure that this paper proves the benefit of this technique.

If possible, I would include similar data for a group of patients who had similar tumors without having angiography performed. I also am not sure that the survival outcomes are helpful in this context, so I would favor not presenting that information. The long term neurlogic status is more important in this study.

Author Response

Thank you very much for your review of our study. We totally agree with your suggestions about our work but, albeit small, we think it could suggest the use of spinal angiography to improve the management of patients with neuroblastoma. We agree that this study cannot demonstrate how angiography directly affects patient survival due to the small sample size and different risk categories, but we are convinced that this technique can improve neurological outcomes and that allowing the surgeon to perform extensive (but still safe) procedures can also indirectly improve the prognosis.

Unfortunately, we don’t have a comparable group of patients at our Institution with the same oncological features who did not undergo angiography; however, we hope to have the chance in the future to make multicenter study to demonstrate, with larger and more variable population, the data presented in this paper.

Round 2

Reviewer 1 Report

Add in the text at the beginning of the results: Ninety-two patients with Neuroblastoma were treated between November 2015 and February 2022 in our Institution, but only six met the eligibility criteria (Thoracic Neuroblastoma with an extension in the costovertebral junction and POSA performed).

Add in the text where you talk about follow-up: The patients were followed up radiologically and clinically, every 2-4 weeks.

Author Response

Thank you for your suggestions! We added the above-mentioned sentences in the text.

Reviewer 3 Report

The manuscript is much improved.

As reviewer there is still two reservations:

1) This is a highly selected group of patients that are being presented. If the authors are to publish in a general paediatric journal this has to be stated emphatically. 

2) This procedure is only achievable in high resourced settings. Although alternatives are discussed in the manuscript, it is important to state that the procedure may not be feasable in lower resourced settings, although the sensitivity is greater for this procedure compare to CT angio. 

Author Response

Thank you for your comments! We corrected the manuscript according to your suggestions, underlying the peculiar characteristics of the analyzed population and the difficulties to proceed with angiography in low-grade settings. 

Reviewer 4 Report

I appreciate the acknowledgement of the small sample size and the limitations that this brings to the study. However, I understand that this is a technique that has not been used for neuroblastoma surgery and may add to the safety of surgical resection of thoracic tumors. 

It is reasonable to publish this report as the English language has been improved and discussion of the limitations has been added - identifying the need to do a larger study to truly assess the benefit of an additional angiogram being performed pre-operatively.

Author Response

Dear Editor, 

Thank you for your appreciation for our study, for your precious suggestion to change the article type, and for your help with these typographical errors.

All the outlined errors have been corrected in the revised manuscript that we uploaded.

We agree with the proposed change in article type in "brief report". Our manuscript's text fits in the category of brief report (I cite from your guidelines, "...The structure is similar to that of an article, and there is a suggested minimum word count of 2500 words..." (https://www.mdpi.com/about/article_types). The online word count of our work is actually 2679.

Can you modify the article type to a brief report for us, or, otherwise, if we have to do it, can you show us how to
modify it online?

Thank you 

Best wishes 
Cristina Martucci
